# Quantitative proteomic analysis of extracellular vesicles in response to baculovirus infection of a *Trichoplusia ni* cell line

Christina Sophie Hausjell[1], Wolfgang Ernst[1], Clemens Grünwald-Gruber[2], Elsa Arcalis[3], Reingard Grabherr[1]*

**1** Department of Biotechnology, Institute of Molecular Biotechnology, University of Natural Resources and Life Sciences, Vienna, Austria, **2** Core Facility Mass Spectrometry, University of Natural Resources and Life Sciences, Vienna, Austria, **3** Department of Applied Genetics and Cell Biology, Institute of Plant Biotechnology and Cell Biology, University of Natural Resources and Life Sciences, Vienna, Austria

* reingard.grabherr@boku.ac.at

**Data Availability Statement:** All relevant data are within the paper and its Supporting Information files.

## Abstract

Due to its outstanding suitability to produce complex biopharmaceutical products including virus-like particles and subunit vaccines, the baculovirus/insect cell expression system has developed into a highly popular production platform in the biotechnological industry. For high productivity, virus-cell communication and an efficient spreading of the viral infection are crucial, and, in this context, extracellular vesicles (EVs) might play a significant role. EVs are small particles, utilized by cells to transfer biologically active compounds such as proteins, lipids as well as nucleic acids to recipient cells for intracellular communication. Studies in mammalian cells showed that the release of EVs is altered in response to infection with many viruses, ultimately either limiting or fostering infection spreading. In this study we isolated and characterized EVs, from both uninfected and baculovirus infected *Tnms42* insect cells. Via quantitative proteomic analysis we identified more than 3000 *T. ni* proteins in *Tnms42* cell derived EVs, of which more than 400 were significantly differentially abundant upon baculovirus infection. Subsequent gene set enrichment analysis revealed a depletion of proteins related to the extracellular matrix in EVs from infected cultures. Our findings show a significant change of EV protein cargo upon baculovirus infection, suggesting a major role of EVs as stress markers. Our study might serve in designing new tools for process monitoring and control to further improve biopharmaceutical production within the baculovirus/insect cell expression system.

## Introduction

The baculovirus/insect cell expression system represents a flexible production platform that offers low cost and high yield production as compared to mammalian cells. Advantageous characteristics such as the ability to perform proper protein folding and comprehensive post-translational modifications, the possibility of cultivation in serum-free media as well as little

**Funding:** This work was funded by the Austrian Science Fund (FWF Project W1224—Doctoral Program on Biomolecular Technology of Proteins —BioToP).

**Competing interests:** The authors have declared that no competing interests exist.

safety concerns are among the reasons why the platform is increasingly used in the biopharmaceutical industry [1]. Licenced vaccines FluBlok, Cervarix and Nuvaxovid, produced in *Spodoptera frugiperda* or *Trichoplusia ni* insect cells, exemplify its outstanding suitability to produce virus-like particles (VLPs) and subunit vaccines [2–4].

Extracellular vesicles (EVs) are small particles released by virtually all eukaryotic cells. They were originally considered a waste product, however, have been reclassified as evidence progressively demonstrated their important role in intracellular communication [5]. The cargo of EVs comprises biologically active compounds such as proteins, lipids as well as nucleic acids including mRNAs and miRNAs. Once EVs are released by one cell, they can be taken up by others and thereby significantly alter state and behaviour of recipient cells [6]. EVs secreted by living cells are generally divided into two main categories, based on size and route of generation: Microvesicles directly bud from the plasma membrane and show a heterogenous size distribution ranging from 100 nm up to 1 μm. Exosomes, with sizes ranging from 30 to 150 nm, are generated from organelles of the endocytotic pathway via invagination of intraluminal vesicles into multivesicular bodies, which ultimately fuse with the plasma membrane and thereby free exosomes into the exterior [7].

Virus-cell interaction is a crucial step in the establishment of efficient viral infections and viruses are known to manipulate and exploit nearly any host cell process to accomplish effective replication and spreading [8]. It is therefore not surprising that various recent studies highlighted that also the release of EVs is altered in response to infection with many viruses including HIV, Influenza, HBV or Zika [9–12]. Viruses hijack EVs to suppress cellular immune responses or increase susceptibility to viral infection by altering the EV host cell protein profile or by purposefully loading viral proteins and nucleic acids into EVs. EBV for instance transfers miRNAs repressing the generation of immunoregulatory compounds such as *CXCL11/ITAC* via EVs resulting in reduced antiviral activity and potentially contributing to persistent infection and HIV packages Nef into exosomes and thereby transports it to resting CD4$^+$ T-cells rendering them permissive [13, 14]. However, EVs might also be utilized by host cells to limit virus spreading, as it was shown that infected cells load immunomodulating molecules into EVs and thereby transfer them to bystander cells provoking an antiviral state. HSV1-infected cells for example deliver STING via EVs to other cells thereby triggering an innate immune response and impeding virus replication [15].

Although, EVs are released by essentially any cell type and therefore, are most likely present in many cell-based production systems, little attention has been drawn to EVs in the context of biotechnological production. Exceptions are several recent studies, which pointed towards a beneficial effect of EVs during CHO cell production processes [16–18]. Besides a possible general impact of EVs on manufacturing processes, within the baculovirus/insect cell expression system EVs might play an additional role, as the system is exceptional among commonly applied biotechnological production platforms in the sense that recombinant production is based on infection of cells by a lytic virus. During infection the baculovirus takes complete control over the host cell, hence the question arises whether the infection also changes the release of EVs and how this might influence infection spreading and thereby finally also the efficiency of the production process [19, 20]. Additionally, when producing secreted products such as VLPs due to similar characteristics regarding size and density EVs and VLPs are difficult to separate during downstream processing, as so an extensive characterization of those co-released vesicles is necessary to develop efficient purification methods [21, 22].

In this study we isolated and characterized EVs from *Tnms42* cells [20, 23]. We utilized a *Trichoplusia ni* cell line, since *T. ni* derived cell lines have been shown to be superior in terms of recombinant protein yield as compared to *Sf9* cells and thus increasingly used in industrial applications [24, 25]. *Tnms42* cells are a derivative of HighFive cells, devoid of latent nodavirus

infection and consequently anticipated to show enhanced robustness and stability during recombinant protein production within the baculovirus/insect cell expression system [26]. To uncover alterations regarding EV release upon infection by baculovirus, cells were cultured uninfected and infected in parallel. 48 h post infection we confirmed strong EV secretion via electron microscopy and western blotting. Subsequent quantitative proteomic analysis and bioinformatic characterization revealed substantial changes in EV protein cargo upon baculovirus infection including a depletion of proteins related to the extracellular matrix (ECM).

## Materials and methods

### Cell culture

*Sf9* cells (ATCC CRL17-11) and *Tnms42* cells (a kind gift from G. Blissard, Boyce Thompson Institute, Ithaca, NY) were cultured in suspension in HyClone SFM4Insect cell culture medium (Cytiva) in shaker flasks at 27˚C and 110 rpm.

### Generation of recombinant baculovirus

The empty vector recombinant baculovirus was generated by transforming MAX Efficiency DH10Bac Competent Cells (Invitrogen) with an empty pACEBac1 plasmid (Geneva Biotech) and subsequently transfecting *Sf9* cells with the resulting bacmid DNA utilizing FuGene HD transfection reagent (Promega) according to the instructions provided by the manufacturer. The virus was amplified to a passage three working stock in *Sf9* cells, and the viral titer was determined by standard plaque assay. Briefly, *Sf9* cells, in 6-well plates, were infected with 10-fold serial dilutions of viral working stocks. After 1 h incubation at 27˚C, the inoculum was removed, and cells were overlayed with HyClone SFM4Insect cell culture medium containing 1% low-gelling temperature agarose (Sigma-Aldrich), 1 x Antibiotic-Antimycotic (Gibco) and 10% FBS (Gibco). Plates were incubated for 6 days at 27˚C in a humid environment, before wells were stained with 1 mg/mL MTT (Sigma-Aldrich), plaques counted, and the titer calculated as plaque forming units/mL.

### Infection and isolation of extracellular vesicles

*Tnms42* cells, grown to mid-log phase, were pelleted at 500 x g for 5 min and taken up in 120 mL fresh culture medium to reach a density of $1 \times 10^6$ cells/mL. Cells were either left uninfected or infected with the empty vector recombinant baculovirus at a multiplicity of infection (MOI) of 5 and incubated for 1 h at 27˚C and 110 rpm. Afterwards, to remove the inoculum, cells were centrifuged at 500 x g for 5 min, pellets were washed, before they were taken up in 120 mL fresh culture medium. Cells were allowed to secrete EVs for 48 h. EVs were then isolated via differential centrifugation. Suspensions were first spun at 300 x g for 5 min, to remove cells, supernatants were further centrifugation at 2,000 x g for 20 min, to remove cell debris followed by a centrifugation step at 10,000 x g for 30 min to eliminate large vesicles. Supernatants were further clarified by filtration using 0.2 μm PVDF-syringe filters. EVs were then isolated and partially purified via ultracentrifugation over a 30% sucrose cushion at 100,000 x g for 2 h at 4˚C using an SW32 Ti rotor (Beckman). To pellet EVs, sucrose cushions were collected, suspended in PBS and again centrifuged at 100,000 x g for 2 h at 4˚C using an SW32 Ti rotor. EV containing pellets were taken up in 10 mM HEPES (Electron microscopy) or PBS (all other analysis). Samples were either directly used for analysis or they were aliquoted, snap frozen and stored at -80˚C.

## Transmission electron microscopy (TEM)

To visualize EVs via electron microscopy, formvar coated copper grids were floated on a 10 μL sample drop for 5 min. After gently removing the excess of sample with whatman paper, grids were quickly washed with distilled water and subsequently floated in a drop of 2% glutaraldehyde for 10 min for fixation. Following a quick wash with distilled water, negative staining was performed with uranyl acetate replacement stain (UAR_EMS Stain, EMS-Electron Microscopy Sciences) by floating the grids on a drop of UAR twice for 10 sec and a third time for 60 sec, gently blotting off the excess of stain after each step. Grids were let air dry prior to imaging using a FEI Tecnai G2 transmission electron microscope operating at 160 kV.

## SDS-PAGE and Western blotting

Protein concentration of EV preparations was determined by Bradford assay utilizing a Coomassie blue G-250-based protein dye reagent (Bio-Rad) and a bovine serum albumin (BSA) standard curve. Samples were mixed with 4 x NuPAGE LDS sample buffer (Invitrogen) supplemented with NuPAGE sample reducing agent (Invitrogen) and heated to 70˚C for 10 min. Equal amounts of protein were separated via SDS-PAGE using NuPAGE 4–12% Bis-Tris mini gels (Invitrogen) and MES SDS running buffer supplemented with NuPAGE antioxidant (Invitrogen). Following SDS-PAGE proteins were electroblotted onto PVDF western blotting membranes (Roche). Membranes were blocked overnight at 4˚C with 3% BSA in PBS-T. Membranes were then incubated with custom-made primary antibodies specific to *T. ni* Hsp70 or Hsp90 (Proteogenix) diluted 1:1000, for 1 h at room temperature. 1:5000 diluted Anti-Rabbit IgG alkaline phosphatase antibody (A3687, Sigma-Aldrich) served as secondary antibody and membranes were again incubated for 1 h at room temperature, before they were developed, and bands visualized with NBT/BCIP (Promega).

## Proteomic analysis

Proteomic analysis was performed with three biological replicates of EVs isolated from uninfected and infected cells resulting in six samples in total. The samples were digested in-solution. The proteins were S-alkylated with iodoacetamide and digested with Trypsin (Promega). The samples were labelled with a TMT 6-plex kit (Thermo Scientific) for quantification according to the manufacturer's protocol and pre-fractionated using high-pH RP LC in 12 fractions.

The fractions were loaded on a nanoEase C18 column (nanoEase M/Z HSS T3 Column, 100Å, 1.8 μm, 300 μm X 150 mm, Waters) using 0.1% formic acid as the aqueous solvent. A gradient from 1% B (B: 80% Acetonitrile, 0.1% FA) to 40% B in 50 min was applied, followed by a 10 min gradient from 40% B to 95% B that facilitates elution of large peptides, at a flow rate of 6 μL/min. Detection was performed with an Orbitap MS (Exploris 480, Thermo) equipped with the standard H-ESI source in positive ion, DDA mode (= switching to MSMS mode for eluting peaks). MS-scans were recorded (range: 350–1200 Da) and a DDA cycle time of 2 sec was used (Isolation width 0.7 m/z). Instrument calibration was performed using Pierce FlexMix Calibration Solution (Thermo Scientific).

The analysis files were analysed using PEAKS, which is suitable for performing MS/MS ion searches [27]. The files were searched against a *T. ni* database downloaded from UniProt (https://www.uniprot.org). For quantification, only proteins with at least 2 peptides were included in further analysis and proteins with a significance higher than 15 were considered as differentially abundant.

## Bioinformatic analysis

To find similarities in protein cargo to mammalian derived EVs, the list of top 100 proteins that are often identified in EVs was downloaded from Vesiclepedia (http://www.microvesicles.org). Gene symbols were converted to human RefSeq protein IDs with the DAVID Gene ID Conversion Tool [28]. A BLAST search was conducted to find *T. ni* homologs for converted human RefSeq protein IDs and found homologs were then manually compared to proteins identified in our proteomics analysis.

To retrieve cellular functions of proteins found in EVs from *Tnms42* cells KEGG pathway mapping was performed with the KEGG Mapper online tool (https://www.genome.jp/kegg).

To gain insight in biological functions of differentially abundant proteins between uninfected and infected samples gene set enrichment analysis (GSEA) was carried out [29]. First *T. ni* RefSeq protein IDs were converted to *D. melanogaster* RefSeq protein IDs via a BLAST search. GSEA was then performed with the top 100 most significantly differentially abundant proteins in infected compared to uninfected EV samples with the WebGestalt toolkit [30]. Gene Ontology (GO) and KEGG pathway were selected as functional databases. The false discovery rate (FDR) was applied as multiple test correction method and the threshold for significance was set to FDR $\leq 0.05$.

# Results and discussion

## Isolation and physical characterization of EVs released by *Tnms42* cells

While most research regarding EVs is focused on mammalian cells, EV-like particles from various other sources, including plants, fungi, parasites, birds, and insects, have also been described [31–35]. Based on previous observations in cell culture supernatants of *Tnms42* cells, which showed the presence of membrane enclosed particles that resemble the structure and size of exosomes and microvesicles from mammalian cells, we hypothesized that also *Tnms42* cells release EV-like vesicles and that the content of those vesicles might be altered in response to baculovirus infection.

To test our hypothesis, suspensions of *Tnms42* cells were infected with an empty backbone recombinant baculovirus at MOI 5 or left uninfected and cultured in parallel for 48 h. For the isolation of EVs, we employed and refined a protocol that has been extensively used for the preparation of EVs from mammalian sources [36, 37]. As outlined in Fig 1 several rounds of centrifugation eliminated cells, debris, and large vesicles. Following a filtration step, that further clarified the conditioned media, EVs were caught in a sucrose cushion and pelleted via another round of ultracentrifugation.

Subsequent TEM analysis confirmed the isolation of substantial amounts of vesicular structures from uninfected as well as infected culture supernatants (Fig 2A, 2B). TEM further showed a heterogeneous population of vesicles with sizes ranging from <40 to <100 nm, comparable to their mammalian equivalents. Additionally, EVs appeared cup-shaped with a depression in their centre, which has also been reported previously as typical for negatively stained EVs [38]. Moreover, we observed higher abundance of EVs in samples from infected cells. Studies in mammalian cells revealed that viral infections tend not only to impact the composition of EVs but might also influence the amount of released vesicles [39].

To further characterize *Tnms42* EVs and to investigate their composition, we performed Western blot analysis. Heat shock proteins such as Hsp70 and Hsp90 are known to be enriched in extracellular vesicles from higher species and are frequently used as EV marker proteins [40]. We, therefore, subjected our samples to Western blot analysis using antibodies against *T. ni* orthologs of Hsp70 and Hsp90. Both heat shock proteins were detected in samples from

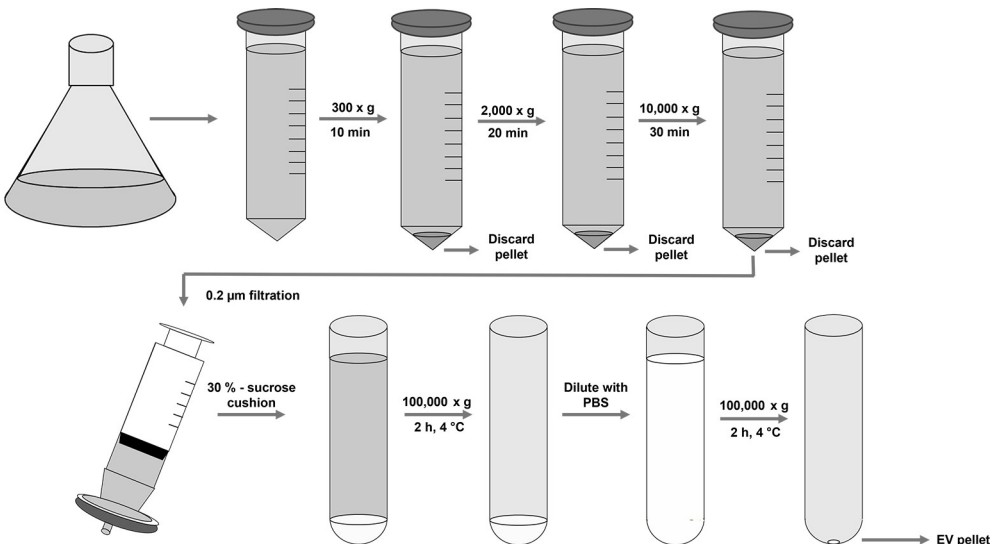

**Fig 1. Schematic overview of the employed EV isolation protocol.** *Tnms42* cells were grown uninfected and infected with baculovirus at MOI 5 in parallel. EVs were isolated from conditioned media 48 hpi. Several low-speed centrifugation steps eliminated cells, cell debris and large vesicles. A filtration step was applied to additionally clarify the conditioned media. Via ultracentrifugation EVs were first caught in a sucrose cushion, washed, and pelleted via another ultracentrifugation step.

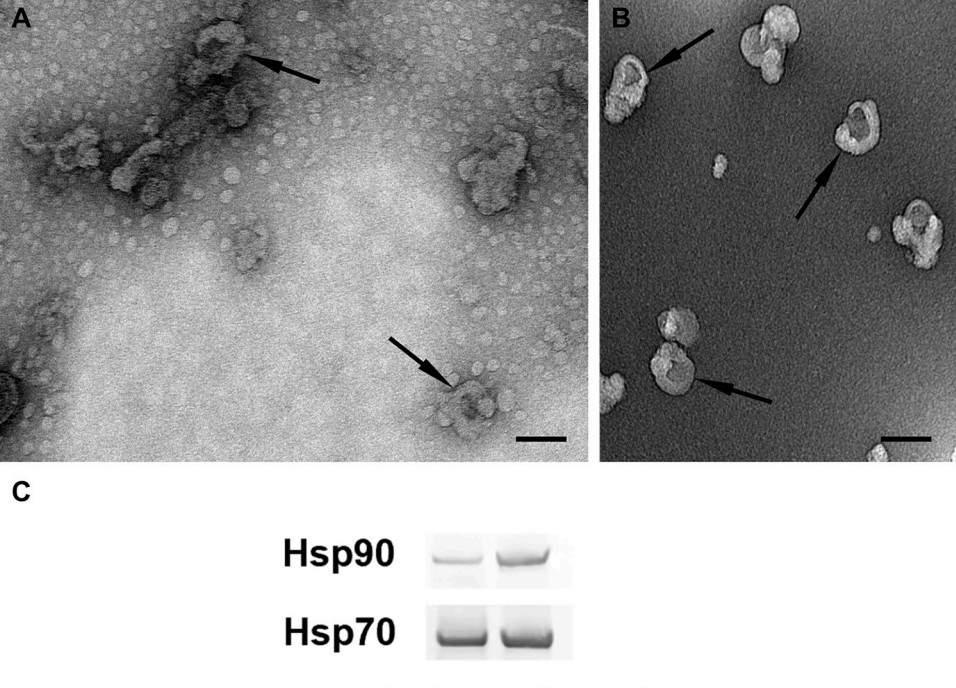

**Fig 2. Physical characterization of *Tnms42* cell EVs.** TEM pictures of negative stained EVs from uninfected (A) and infected (B) cells showed cup-shaped vesicles (arrows). Bars 100 nm. Western blot analysis (C) revealed incorporation of EV marker proteins Hsp90 and Hsp70 into *Tnms42* cell EVs and an enrichment of Hsp90 in EVs upon baculovirus infection.

uninfected as well as infected cells, further verifying successful EV isolation (Fig 2C). For Hsp90 a substantially stronger band appeared for samples isolated from infected cells, indicating an enrichment of Hsp90 in EVs upon baculovirus infection.

## Quantitative proteomic analysis reveals alterations in EV protein cargo upon baculovirus infection

To deepen the insight into protein composition of *Tnms42* cell-EVs and to identify possible changes upon virus infection, we conducted quantitative proteomic analysis. Isobaric labelling via Tandem Mass Tags (TMT) is a high-throughput technique that allows for relative protein quantification of different samples during a single round of MS analysis [41]. We, therefore, performed TMT labelled LC-ESI-MS/MS with a total of six samples, resulting from EVs isolated from uninfected and infected cultures with three biological replicates of each group.

In total 3171 *T. ni* proteins were identified in *Tnms42* EVs, originating from the two samples and their three replicates. As visualized in Fig 3A, of the 3171 identified proteins, 402 were differently abundant in EVs from infected and uninfected cells (significance >15). Whereof, 247 were significantly enriched and 155 significantly depleted in EVs released by baculovirus infected cells, respectively. Interestingly, Hsp90 was amongst the 247 significantly enriched proteins, confirming the results obtained by western blotting. All identification and quantification results can be found in the S1 File.

We did not include viral proteins in our search, since the employed isolation protocol does not explicitly distinguish between EVs and baculoviral particles and we therefore cannot exclude that baculoviral proteins originate from co-purified viral particles instead of an incorporation into EVs. We are aware of this limitation regarding our study, however due to an overlap in size and density it was not possible to completely separate both particle populations.

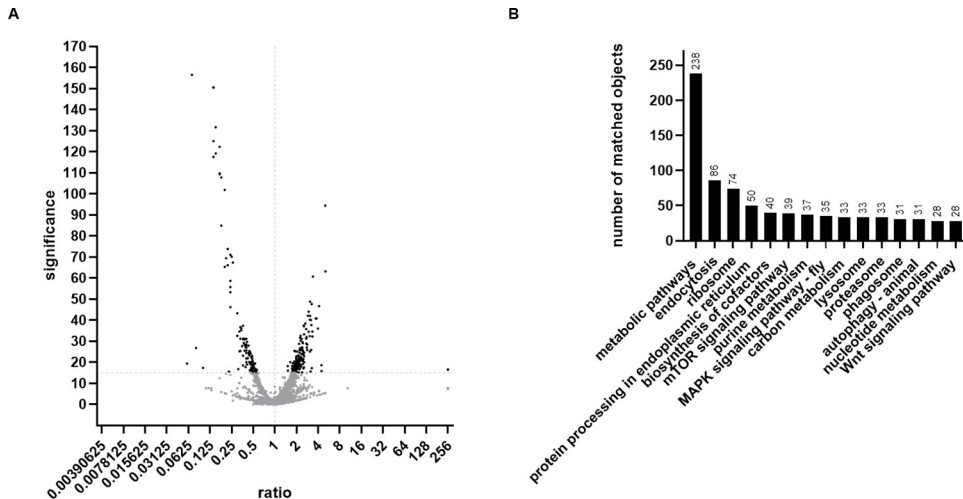

**Fig 3. Proteomic analysis of EVs from uninfected and baculovirus infected *Tnms42* cells.** (A) The volcano plot shows differentially abundant proteins in EVs from infected compared to uninfected cells. The horizontal dashed grey line represents the set significance threshold (> 15) and significantly differentially abundant proteins are highlighted in black. (B) The bar chart shows the KEGG pathway analysis and the number of matched objects for the top 15 categories with the most matched objects, including metabolic pathways, endocytosis and signalling pathways.

## *Tnms42* EVs show substantial similarities to EVs from higher species

To shed light on cellular processes governing EV release in *Tnms42* cells, to point out functional characteristics of their EV-protein cargo as well as to identify alterations thereof induced by baculovirus infection, we performed several bioinformatic characterizations.

Since this is to our knowledge the first study investigating *Trichoplusia ni* cell derived EVs we first aimed to detect possible similarities in protein composition to well characterized EVs from higher species. We therefore ran a BLAST search to find homologs of the top 100 most often identified proteins in mammalian derived EVs as deposited in the Vesiclepedia database. We found homologs for 88% of the proteins and a majority of them (64) was also present in our proteomics data, amongst them tetraspanins (CD9, CD81), heat shock proteins (HSPA8, HSP90AA1), annexins (ANXA5, ANXA6), cytoskeleton elements (ACTB) as well as enzymes (GAPDH, PKM, PGK1) and ESCRT components (PDCD6IP, TSG101) (S2 File).

We then subjected all identified proteins to KEGG pathway analysis using the KEGG Mapper online tool. The pathways with the most matched objects were by far metabolic pathways (238) followed by endocytosis (86) and ribosome (74). Additionally, the signalling pathways, mTOR signalling pathway (39) and MAPK signalling pathway (35) also contained a notable number of objects (Fig 3B). The main function of EVs is intracellular communication and uptake of EVs can result in considerable alterations of intrinsic cellular patterns, it is therefore not surprising that the cargo of EVs includes high quantities of metabolic enzymes and proteins involved in signalling. Further, the fact that endocytosis is one of the pathways with the most matched objects might be due to the fact that organelles of the endocytotic pathway are the main site of exosome biogenesis [42]. A list of all classifications can be found in the S3 File.

Collectively, our analysis shows that *Tnms42* insect cell-EVs possess high similarity to their mammalian analogous, which is in accordance with previous studies indicating that vesicle-based intracellular communication is a highly conserved process [43, 44].

## EVs from baculovirus infected cells are depleted in extracellular matrix proteins

To better understand underlying biological functions of changes in EV protein composition following baculovirus infection we performed gene set enrichment analysis. Most popular online tools for GSEA are only available for model organisms and thus do not contain data for *Trichoplusia ni*. We therefore converted *T. ni* protein IDs to *Drosophila melanogaster* IDs via a BLAST search. By doing so, we found homologs for 89% of the identified *T. ni* proteins. We then subjected the top 100 differentially abundant proteins between uninfected and infected samples to GSEA using the WebGestalt online tool. GSEA failed to identify groups that were significantly enriched in one of the two experimental set-ups for the Gene Ontology categories biological process and molecular function. However, the analysis for the GO category cellular component showed that the term extracellular matrix was significantly depleted in EVs from infected cells compared to uninfected samples (Fig 4A). In agreement thereto, when performing GSEA with KEGG pathway as functional database, the term ECM-receptor interaction was significantly depleted in EVs from infected in comparison to uninfected cells (Fig 4B). Besides its well-recognized purpose of providing structural stability in tissue, the ECM is involved in many physiological processes including cell growth, migration, apoptosis, and signalling. To be able to execute its diverse functions the ECM is highly dynamic, and its architecture constantly undergoes massive rearrangements [45]. It was previously shown that EVs might play a significant role in the context of ECM remodelling, as they contain extracellular matrix components which might be deposited, or which are involved in signalling and hampered EV release can massively change ECM arrangements [46]. In their native environment

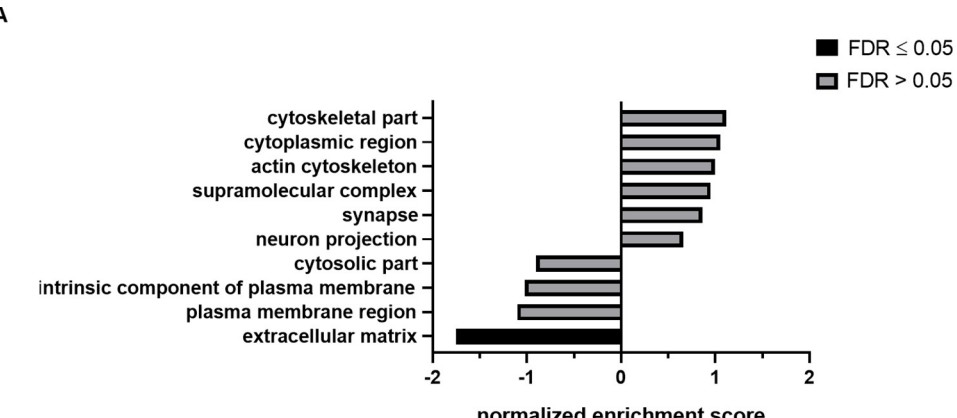

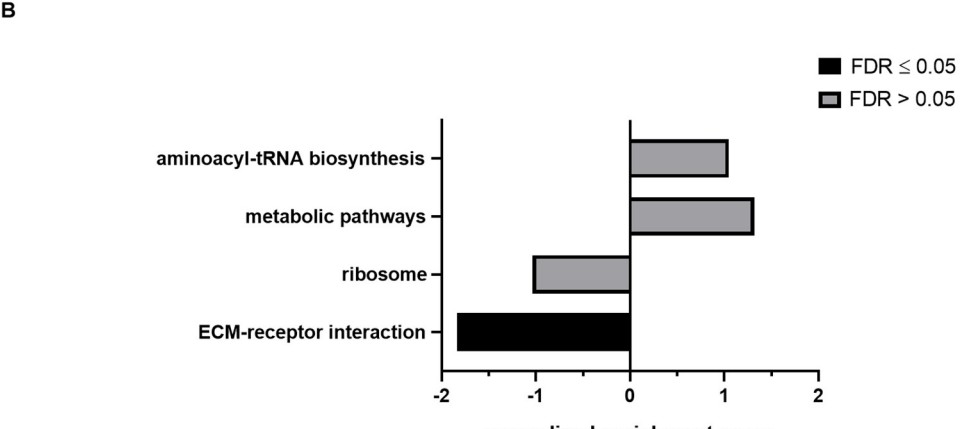

**Fig 4. GSEA of the top 100 most significantly differentially abundant proteins.** (A) The graph shows enriched terms upon baculovirus infection for the GO category cellular component with a significant depletion of the term ECM. (B) The bar chart presents enriched categories upon baculovirus infection utilizing KEGG pathway as functional database and a significant depletion of the term ECM-receptor interaction (FDR < 0.05).

baculoviruses enter insects via ingestion and in order to establish systemic infection in the insect larvae they must escape the midgut which requires surmounting permeability barriers including the basal lamina, a thin layer of ECM [47]. Lauko *et al.* showed that baculovirus infection induces the formation of invadosome clusters potentially triggering an active degradation of ECM and Means and Passarelli reported a signalling cascade ultimately leading to rearrangement of the basal lamina upon baculovirus infection [48, 49]. It is therefore tempting to speculate that baculoviruses actively downregulate the packaging of ECM components into EVs and thereby counteract the assembly and renewal of ECM layers, which might otherwise hamper systemic infection.

Aside from our findings indicating a potential role of EVs in systemic infection following oral uptake of baculovirus by insects, EVs might also be of relevance during biotechnological production. It was recently shown that EV enrichment in CHO cell cultures inhibits apoptosis, likely having a beneficial effect on product yield [50]. Our study shows substantial differences in EV-release upon baculovirus infection, pointing towards a role of EVs as stress markers within the baculovirus/insect cell expression system. Our findings provide the basis for future studies, such as EV enrichment and depletion, which will show how this alteration might

influence susceptibility to baculoviral infection and thereby ultimately also the product yield during biopharmaceutical production. Further, our characterization of vesicles, that are co-released from cells during production might help in developing efficient purification tools for difficult to separate products such as VLPs.

## Conclusion

In this study extracellular vesicles from *Tnms42* insect cells and changes upon infection with baculovirus were investigated. EVs were isolated from infected and uninfected cultures via differential centrifugation. TEM and western blotting confirmed strong EV secretion by insect cells and revealed resemblances to mammalian equivalents. Subsequent quantitative proteomic analysis determined the presence of more than 3000 *T. ni* proteins in both samples of which more than 400 were differentially abundant upon infection. Subsequent GSEA showed a strong depletion in EV proteins associated with the ECM after infection, potentially serving the baculovirus in passing barriers after ingestion by insects and thereby fostering systemic infection. Together, our data show a marked alteration of EV release upon infection with baculovirus in terms of amount as well as protein cargo. Our findings serve as starting point for future studies which will reveal whether those alterations ultimately also influence spreading of baculovirus infections, which other beneficial or unfavourable properties EVs might bring along for biotechnological production utilizing the baculovirus/insect cell expression system and how they might be used as tool for process monitoring and control.

## Supporting information

**S1 Raw images.**
(PDF)

**S1 File. Protein identification and quantification.**
(XLSX)

**S2 File. Comparison proteins in EVs.**
(XLSX)

**S3 File. KEGG pathway analysis.**
(DOCX)

## Acknowledgments

The MS equipment was kindly provided by the EQ-BOKU VIBT GmbH and the BOKU Core Facility Mass Spectrometry.

## Author Contributions

**Conceptualization:** Christina Sophie Hausjell, Reingard Grabherr.

**Formal analysis:** Clemens Grünwald-Gruber.

**Funding acquisition:** Reingard Grabherr.

**Investigation:** Christina Sophie Hausjell.

**Methodology:** Christina Sophie Hausjell, Clemens Grünwald-Gruber, Elsa Arcalis.

**Project administration:** Reingard Grabherr.

**Supervision:** Wolfgang Ernst, Reingard Grabherr.

**Writing – original draft:** Christina Sophie Hausjell.

**Writing – review & editing:** Wolfgang Ernst, Clemens Grünwald-Gruber, Elsa Arcalis, Reingard Grabherr.

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
