## [Decision Letter · Decision Letter 0]

26 Dec 2022

PONE-D-22-32175Quantitative proteomic analysis of extracellular vesicles in response to baculovirus infection of a Trichoplusia ni cell linePLOS ONE

Dear Dr. Grabherr,

Thank you for submitting your manuscript to PLOS ONE. After careful consideration, we feel that it has merit but does not fully meet PLOS ONE’s publication criteria as it currently stands. Therefore, we invite you to submit a revised version of the manuscript that addresses the points raised during the review process.

We look forward to receiving your revised manuscript.

Kind regards,

Jian Xu, Ph.D.

Academic Editor

PLOS ONE

Journal Requirements:

Reviewers' comments:

Reviewer's Responses to Questions

**Comments to the Author**

1. Is the manuscript technically sound, and do the data support the conclusions?

Reviewer #1: Partly

Reviewer #2: Yes

2. Has the statistical analysis been performed appropriately and rigorously? 

Reviewer #1: Yes

Reviewer #2: Yes

3. Have the authors made all data underlying the findings in their manuscript fully available?

Reviewer #1: Yes

Reviewer #2: Yes

4. Is the manuscript presented in an intelligible fashion and written in standard English?

Reviewer #1: Yes

Reviewer #2: Yes

5. Review Comments to the Author

Reviewer #1: This paper presents for the first time an account of the isolation and proteomic characterisation of EVs from non-infected and baculovirus-infected Tni cells. The study is detailed and results presented suitably. I have just one concern that I think the authors should address prior to publication. EVs and baculovirus budded virus have very similar biophysical properties making it very hard to separate them completely. I can't find any recognition of this problem mentioned. What controls, or what limitations on the data and subsequent conclusions, did the authors put in place to ensure that the EVs isolated from baculovirus-infected cells were not contaminated by BV? Are the authors confident that the data from infected cell EVs was not at least partially attributable to the presence of BV? Did the authors demonstrate that their method to purify EVs avoided co-purification of low levels of BV? Controls may well have been put in place but I couldn't find any mention of these in the current version of the paper. If the authors can't be certain, the text should be modified to address this potential limitation.

Reviewer #2: In this study the authors isolated and characterized EVs from Tnms42 cells, a derivative of High Five cells. These studies are interesting to the field of baculovirus expression vector system (BEVS). The results are solid and with a good scientific writing skill. Two concers should be included in the discussion section:

1. In BEVS, the most popular used cells are Sf9, why the authors used the High Five cells?

2. There are studies employed the microarray or RNAseq to analyze the gene expression profile change after the baculovirus infection in,Sf9 or High Five cells or BmN cells (derived from Bombyx mori). I am interesting to know whether the proteins expression change in EVs reveled in this manuscript also revealed in the previous studies, like (1) Expression of baculovirus genes in permissive and nonpermissive cell lines Biochemical and Biophysical Research Communications Volume 323, Issue 2, 15 October 2004, Pages 599-614. (2) DNA microarrays of baculovirus genomes: differential expression of viral genes in two susceptible insect cell lines J. Yamagishi, R. Isobe, T. Takebuchi & H. Bando Archives of Virology volume 148, pages587–597 (2003). I suggest these studies should be included in the studies to check the consistency.

6. PLOS authors have the option to publish the peer review history of their article (what does this mean?). If published, this will include your full peer review and any attached files.

Reviewer #1: No

Reviewer #2: **Yes**

---

## [Author Response · Author response to Decision Letter 0]

13 Jan 2023

Response to reviewers

Reviewer#1: This paper presents for the first time an account of the isolation and proteomic characterisation of EVs from non-infected and baculovirus-infected Tni cells. The study is detailed and results presented suitably. I have just one concern that I think the authors should address prior to publication. EVs and baculovirus budded virus have very similar biophysical properties making it very hard to separate them completely. I can’t find any recognition of this problem mentioned. What controls, or what limitations on the data and subsequent conclusions, did the authors put in place to ensure that the EVs isolated from baculovirus-infected cells were notcontaminated by BV? Are the authors confident that the data from infected cell EVs was not at least partially attributable to the presence of BV? Did the authors demonstrate that their method to purify EVs avoided co-purification of low levels of BV? Controls may well have been put in place but I couldn’t find any mention of these in the current version of the paper. If the authors can’t be certain, the text should be modified to address this potential limitation.

We agree that a limitation of our study is the co-isolation of baculovirus particles. Due to highly similar characteristics between budded baculovirus and EVs regarding size and density, we were not able to completely separate both populations. We mentioned this limitation in the results and discussion section of the submitted manuscript: page 11-12, line 264-267 (page 12, line 269-272 revised version)- “We did not include viral proteins in our search, since the employed isolation protocol does not explicitly distinguish between EVs and baculoviral particles and we therefore cannot exclude that baculoviral proteins originate from co-purified viral particles instead of an incorporation into EVs.” 

To emphasize this limitation of our study, following sentence was added in the revised manuscript (page 12, line 272-274): “We are aware of this limitation regarding our study, however due to an overlap in size and density it was not possible to completely separate both particle populations.”

Reviewer#2: In this study the authors isolated and characterized EVs from Tnms42 cells, a derivative of High Five cells. These studies are interesting to the field of baculovirus expression vector system (BEVS). The results are solid and with a good scientific writing skill. Two concers should be included in the discussion section:

1. In BEVS, the most popular used cells are Sf9, why the authors used the High Five cells?

2. There are studies employed the microarray or RNAseq to analyzethe gene expression profile change after the baculovirus infection in,Sf9 or High Five cells or BmN cells (derived from Bombyx mori). I am interesting to know whether the proteins expression change in EVs reveled in this manuscript also revealed in the previous studies, like (1) Expression of baculovirus genes in permissive and nonpermissive cell lines Biochemical and Biophysical Research Communications Volume 323, Issue 2, 15 October 2004, Pages 599-614. (2) DNA microarrays of baculovirus genomes: differential expression of viral genes in two susceptible insect cell lines J. Yamagishi, R. Isobe, T. Takebuchi & H. Bando Archives of Virology volume 148, pages587–597 (2003). I suggest these studies should be included in the studies to check the consistency.

1. We agree that Sf9 cells are among the most popular cell lines within the baculovirus/insect cell expression system. However, several studies have shown that Hi5 cells are advantageous in terms of yield for many recombinant proteins (especially secreted proteins) as compared to Sf9 cells and thus, are increasingly applied in biotechnological production. Both cell lines, Sf9 and Hi5, are however carriers of latent viral infections, what might ultimately negatively influence the final product yield. Therefore, new, robust, virus-free insect cell lines are needed and requested by commercial producers. In this study we utilized Tnms42 cells, a derivative of Hi5, free of latent nodavirus infection and therefore of interest for future industrial application. The information was added and the text in the introduction section page 4, line 79-87 changed accordingly, including three new references ([24-26] see below). 

“In this study we isolated and characterized EVs from Tnms42 cells [20, 23]. We utilized a Trichoplusia ni cell line, since T. ni derived cell lines have been shown to be superior in terms of recombinant protein yield as compared to Sf9 cells and thus increasingly used in industrial applications [24, 25]. Tnms42 cells are a derivative of HighFive cells, devoid of latent nodavirus infection and consequently anticipated to show enhanced robustness and stability during recombinant protein production within the baculovirus/insect cell expression system [26]. To uncover alterations regarding EV release upon infection by baculovirus, cells were cultured uninfected and infected in parallel.”

24. Krammer F, Schinko T, Palmberger D, Tauer C, Messner P, Grabherr R. Trichoplusia ni cells (High Five) are highly efficient for the production of influenza A virus-like particles: a comparison of two insect cell lines as production platforms for influenza vaccines. Mol Biotechnol. 2010;45(3):226-34. Epub 2010/03/20. doi: 10.1007/s12033-010-9268-3. PubMed PMID: 20300881; PubMed Central PMCID: PMCPMC4388404.

25. Palmberger D, Wilson IB, Berger I, Grabherr R, Rendic D. SweetBac: a new approach for the production of mammalianised glycoproteins in insect cells. PLoS One. 2012;7(4):e34226. Epub 2012/04/10. doi: 10.1371/journal.pone.0034226. PubMed PMID: 22485160; PubMed Central PMCID: PMCPMC3317771.

26. Wilde M, Klausberger M, Palmberger D, Ernst W, Grabherr R. Tnao38, high five and Sf9--evaluation of host-virus interactions in three different insect cell lines: baculovirus production and recombinant protein expression. Biotechnol Lett. 2014;36(4):743-9. Epub 2014/01/01. doi: 10.1007/s10529-013-1429-6. PubMed PMID: 24375231; PubMed Central PMCID: PMCPMC3955137.

2. Such a comparison would provide a valuable contribution, however, is beyond the objective of the presented work, since the aim of the study was to investigate changes in host cell protein composition of EVs upon baculovirus infection. As stated on page 11-12, line 264-267 (page 12, line 269-272 revised version), we did not include viral proteins in our analysis, since the applied purification protocol does not explicitly distinguish between baculovirus particles and extracellular vesicles and we therefore cannot exclude the possibility that viral proteins originate from viral particles instead of an incorporation into EVs. Since we investigated alterations in the host cell protein composition and did not have a look at viral proteins in EVs we unfortunately cannot compare our proteomics data to the mentioned publications. Moreover, it was not our aim to compare several cell lines, instead we investigated the change in proteome of one chosen cell line. However there exists a transcriptome study, comparing non-infected and infected Tnms42 cells (Koczka, et al., 2018. doi: 10.1016/j.jbiotec.2018.02.001).

---

## [Editor Report · Decision Letter 1]

17 Jan 2023

Quantitative proteomic analysis of extracellular vesicles in response to baculovirus infection of a Trichoplusia ni cell line

PONE-D-22-32175R1

Dear Dr. Grabherr,

We’re pleased to inform you that your manuscript has been judged scientifically suitable for publication and will be formally accepted for publication once it meets all outstanding technical requirements.

Kind regards,

Jian Xu, Ph.D.

Academic Editor

PLOS ONE
---

## [Editor Report · Acceptance letter]

20 Jan 2023

PONE-D-22-32175R1 

Quantitative proteomic analysis of extracellular vesicles in response to baculovirus infection of a *Trichoplusia ni* cell line 

Dear Dr. Grabherr:

I'm pleased to inform you that your manuscript has been deemed suitable for publication in PLOS ONE. Congratulations! Your manuscript is now with our production department. 

Kind regards, 

on behalf of

Dr. Jian Xu 

Academic Editor

PLOS ONE